# Barriers and Facilitators to Accessing Health Services: A Qualitative Study Amongst People with Disabilities in Cameroon and India

**DOI:** 10.3390/ijerph16071126

**Published:** 2019-03-29

**Authors:** Maria Zuurmond, Islay Mactaggart, Nanda Kannuri, Gudlavalleti Murthy, Joseph Enyegue Oye, Sarah Polack

**Affiliations:** 1Department of Clinical Research, Faculty of Infectious and Tropical Diseases, London School of Hygiene & Tropical Medicine, London WC1E 7HT, UK; islay.mactaggart@lshtm.ac.uk (I.M.); GVS.Murthy@lshtm.ac.uk (G.M.); sarah.polack@lshtm.ac.uk (S.P.); 2Indian Institute of Public Health, Hyderabad 122002, India; nandu.k@iiphh.org; 3Sightsavers Yaounde, Cameroon; joye@sightsavers.org

**Keywords:** disability, health services, barriers, Cameroon, India

## Abstract

*Background*: Article 25 of the UNCRPD stipulates the right of people with disabilities to the highest attainable standard of health, and the timely uptake of appropriate health and rehabilitation services. This study seeks to explore the factors which influence access to health care among adults with disabilities in Cameroon and India. *Methods*: A total of 61 semi-structured interviews were conducted with a purposive sample of adults with vision, hearing or musculoskeletal impairments, using data from an earlier cross-sectional disability survey. In addition, 30 key informants were interviewed to provide contextual information about the local services and context. *Results*: Key themes included individual-level factors, understanding and beliefs about an impairment, and the nature of the impairment and interaction with environmental factors. At the community and household level, key themes were family dynamics and attitudes, economic factors, social inclusion and community participation. Intersectionality with gender and age were cross-cutting themes. Trust and acceptability of health service providers in India and poor understanding of referral processes in both countries were key service-level themes. *Conclusions*: The interaction of environmental and personal factors with the impairment and their levels of participation and inclusion in community structures, all contributed to the take up of services. This study illustrated the need for a multi-faceted response to improve access to health services for people with disabilities.

## 1. Background

There are an estimated 1 billion people in the world living with disabilities [1], the majority of whom (80%) live in low and middle-income settings. The United Nations Convention on the Rights of Persons with Disabilities (UNCRPD) defines people with disabilities as those with “long-term physical, mental, intellectual or sensory impairments which, in interaction with various barriers, may hinder their full and effective participation in society on an equal basis with others” [2] People with disabilities are disproportionately over-represented among the most marginalised in society; evidence suggests that people with disabilities on average have lower educational attainment, poorer health, lower economic opportunities and are at increased risk of poverty [1].

Evidence suggests people with disabilities, on average, experience poorer health and, therefore, have greater general health needs compared to others in the population [3,4]. This may be for several reasons; they are more likely to experience socioeconomic disadvantage and are therefore at increased risk of conditions associated with poverty and underlying health conditions or impairments that may lead to other health consequences (for example spinal cord injuries increase the vulnerability to pressure sores). At the same time, some people with disabilities also have additional healthcare needs related to their underlying health condition or impairment such as specialist ophthalmic or audiometric interventions, or physical rehabilitation. These specialised interventions can be instrumental in facilitating independence, social and economic participation and quality of life [5].

Article 25 of the UNCRPD stipulates the right of people with disabilities to the highest attainable standard of health [2]. In addition, it articulates the right of all people with disabilities, including those in rural areas, to the provision of quality health services available as close as possible to their own communities [2]. Health is a core focus of the Sustainable Development Goals (SDGs) and Goal 3 includes a call to improve access to healthcare services for all through Universal Health Coverage. The SDGs explicitly emphasise ‘leaving no one behind’, with people with disabilities identified as a particularly vulnerable or marginalised group [6] at risk of poor access to health care. It is therefore essential to understand and meet the health and rehabilitation needs of people with disabilities globally if the SDGs are to be achieved.

The timely uptake of appropriate health and rehabilitation services can be important for maximising functioning, participation and improving quality of life. Despite international commitments to access to health care for all, there is evidence to suggest that people with disabilities face a range of physical, financial, communication and attitudinal barriers which limit their access to healthcare services [7]. However, there is limited evidence available exploring in-depth the individual, family and community factors that influence whether adults with disabilities seek health services in Low and Middle-Income Countries (LMICs) [8,9,10,11,12]. Understanding this important for informing the design and implementation of disability-inclusive health care programmes.

To address this gap, this study seeks to explore the factors which influence access to health care among adults with disabilities in Cameroon and India. 

## 2. Methods

### 2.1. Conceptual Framework

In our study, we draw upon two key frameworks to explore decision making related to health care seeking. Firstly, the socio-ecological framework [13,14] outlines the interactive effects of personal and environmental factors at different levels: individual, interpersonal, community and organisational. In this study, the emphasis was at the level of micro-systems; with a focus on the individual and their perspective on their interaction with the family and community. Secondly, we draw upon the World Health Organisation’s International Classification of Functioning, Disability and Health (ICF) framework [15], which conceptualises a person’s level of functioning and disability as a dynamic interaction between their health condition, environmental and personal factors.

### 2.2. Study Settings

In Cameroon, a study was undertaken in Fundong Health District in the North-West region, one of two English-speaking regions in the country with an estimated population size of 125,604. Cameroon is ranked 153 (out of 185) in the United Nations Human Development index [16] with a life expectancy of 58.1 years [17]. The North-west region is a predominately rural (63% of the region) mountainous area, heavily reliant on agriculture and subsistence farming as the main source of livelihoods. There is a strong traditional organisation of chiefdoms. The main ethnic group in Fundong is the ‘Kom’ who have a matrilineal kin system. The main ethnic group in Bamenda is the ‘Kom’ who have a matrilineal kin system. The estimated all-age prevalence of disability is 10.5% [18]. The health system is pyramidal with central, regional and district level services. At each level, there is a mix of government and private health facilities. There are no rehabilitation services in the government health facilities; there is one large faith-based hospital which offers rehabilitation services, and a number of NGOs also provide support, which includes providing assistive devices.

In India, a study was conducted in the Mahabubnagar District, Telangana State, India. India is ranked 131 in the United Nations Human Development index [16] with a life expectancy of 69 years [17]. The majority (85%) of the population in the Mahabubnagar district live in rural areas and approximately 48% are literate [19]. The estimated all-age prevalence of disability is 8.4% [18]. Healthcare is a mixed services model as there is no national health service. There is a socioeconomic gradient with the wealthier people preferring private health care and poorer people more commonly seeking public health systems. Usually, the more deprived areas depend on public health systems, and a significant proportion of people still depend on village level health practitioners. Rehabilitation services are mostly provided by the government systems supported by the NGO sector, whilst the certification and grading of disabilities are done through the government health system.

### 2.3. Study Participants

This study built on earlier large-scale surveys which aimed to assess the prevalence and impact of disabilities in two contrasting LMIC settings; one African and one Asian setting. The methodology and results of these surveys are available in detail elsewhere [3,18]. Briefly, in each setting, approximately 4000 people were sampled using standard sampling methods and assessed for disability through (i) self-reported limitations in functioning using the Washington Group Extended Set on Functioning and (ii) clinical assessment for vision, hearing, musculoskeletal impairments or depression. Participants were considered to have a disability if they reported a lot of difficulties/inabilities functioning in the core functioning domains or if they had moderate or worse vision, hearing, musculoskeletal impairment. Participants identified as having an impairment were referred to accordingly. In Cameroon, they were referred to a large faith-based hospital in the district for free or subsidised services. In India they were referred to medical and rehabilitative services available in the region; a mix of governmental and private services. Although data on referral attendance was not collected, these follow up discussions with health professionals and participants suggested that uptake was likely to be low.

For this qualitative study, we selected adults with vision, hearing or musculoskeletal impairments identified in the survey. Purposive sampling was used to ensure representation from men and women and different age groups, impairment types and distance from the main health services. In total, we selected 30 participants from 12 villages in Cameroon and 31 from 10 villages/Mandals in India. Additionally, 12 key informants in India and 18 key informants in Cameroon were interviewed to provide contextual information about the local services and context. These included representatives from the ministry of health and social service, NGO providers, clinical staff, and representatives from disabled people’s organisations.

Table 1 shows characteristics of the study population. All 30 participants in Cameroon had a moderate or severe vision and hearing or musculoskeletal impairment identified through clinical assessment. In India, 27 participants had moderate or severe impairments and four had mild impairments. In Cameroon 5 people reported having an impairment from birth, 17 reported acquiring their impairment during their adult lifetime and 5 attributed it to ageing. In contrast in India, 14 people identified as having an impairment from birth/early childhood, 8 acquired this during their adult lifetime and 9 attributed it to ageing. The average age of the sample was 51 years in Cameroon and 49 years in India. Overall levels of education were low: 43% with no education, 47% with primary education and 3% with secondary education in Cameroon; and 65% with no education, 13% with primary level education, and 23% with secondary education in India. A total of 67% of the Cameroon sample had worked in the last 12 months and 61% in India.

### 2.4. Data Collection

Semi-structured interviews were conducted approximately 2–3 months after the disability prevalence surveys; in November 2013 in Cameroon and in June 2014 in India. Interviews were conducted by two interviewers at each site: one local and one UK researcher. They lasted approximately one hour and were conducted in the respondent’s homes in the local language of Kom or pidgin English (Cameroon) and in Urdu or Telugu (India). Interviews were recorded and detailed notes were also taken to facilitate the discussion of key issues at the end of each day with translators.

The interviews covered a broad range of questions about participation in family and community life, access to health services, and access to livelihoods. The focus of this article is on access to health services. All interviews were conducted in the home setting and were conducted, to the extent it was possible, in a private setting.

We were unable to conduct direct interviews with five selected participants: in four cases this was because the person had a profound hearing impairment and was unable to use sign language, and in one case the person had an intellectual and communication impairment. For these participants interviews were undertaken with their primary caregiver.

### 2.5. Analysis

Interviews were audio-recorded and transcribed into English. In India a double transcription was conducted; into the local language first and then into English. Interview transcripts were coded by one lead researcher (UK-MZ), and key themes and sub-themes were checked and verified with the local researcher in each country (NK). A thematic analysis of key issues was undertaken using an iterative process of both a priori codes and emergent new themes which emerged from the analysis of the data, in line with good practice for qualitative data analysis [20]. The data were further interrogated against attributes which included the type of impairment, the level of severity of the impairment, the gender, and age. NVivo 10 software (QSR International, Burlington, MA US) was used to manage the data.

### 2.6. Ethics

Ethical Approval for the study was granted by The London School of Hygiene & Tropical Medicine (6207, London, UK), National Ethics Committee for Research in Human Health (2013/03/084/1/CNERSH/SP CNERSH, Cameroon), Cameroon Baptist Convention Health Board Institutional Review Board (2013-07), Indian Institute of Public Health Hyderabad Institutional Ethics Committee and the Government of India Health Ministry Screening Committee.

## 3. Results

During the interviews, the respondent’s experience of accessing general health services and impairment-specific services (e.g., hearing aids for hearing impairment) were often discussed together rather than as separate issues. Further, the factors that influenced the uptake of those impairment-related services that participants were referred to in the survey were interwoven into their narratives about access to health care more generally, as opposed to the two being clearly delineated. Where there was a clear differentiation, we have sought to highlight this, but in practice, their impairment, such as poor vision, was often part of a continuum of health care needs, including other illnesses, with a variety of common and overlapping factors impacting on decisions to seek health care.

### 3.1. Individual Level—Beliefs and Understanding

Individual beliefs about their impairment shaped participant’s decisions to seek care and treatment. In both settings, it was not uncommon for traditional treatments to be sought, alongside, or instead of, biomedical options for treatment. Most commonly, there was poor biomedical understanding about their impairment or underlying health condition, and of treatment options available to them. This, in turn, meant they were less likely to seek biomedical treatment, illustrated here by one woman in her thirties in India, who delayed seeking a diagnosis for poor vision for five years because *“my village people told me that, because my mother and mother-in-law died, I cried a lot, and that is the reason there was more water in my eyes”*. 

Traditional medicine and remedies and private medicine sellers were commonly discussed as a preferred treatment option in the Indian context; for example, one woman described her preference to use either honey or goat’s milk to treat her eyes, in part, because she feared that she might need an operation if she attended the hospital.

The beliefs and understanding of other household members were also influential; any decisions to seek health or rehabilitation services were commonly made at the family level, and not by the individual. In this context, disability-related stigma emerged, sometimes acting as a barrier to the uptake of services:

*The family members of the person who is disabled really act as a big barrier, because they look at them like nothing good can come out of them. They look at them as pythons and ill omens, so they are not willing to spend a dime on those who are disabled.* (KI, Cameroon)

### 3.2. Prioritisation of Competing Demands for Health Care 

Another common reason for delayed health-seeking behaviour related to impairments, for example, the gradual loss of vision, was competition for treatment for other health conditions. Priority was commonly given to treating acute conditions, or conditions perceived to be more urgent. This was most noticeable among older people interviewed in India, who, for example, wanted to treat their diabetes or high blood pressure. In contrast, they were often willing to cope with a decline in vision and hearing, which some participants seemed to have adapted to. In one-third of interviews in Cameroon and half of those in India, there was evidence of a ‘weighing up’ process of competing demands for health care. This was illustrated in India by the case of a 65-year-old woman who had not accessed the free ophthalmic services offered for her severe visual impairment, and yet she was prepared to travel on four buses, for several hours unaccompanied, to have treatment for a kidney condition.

Again, this prioritisation process was not only made at the individual level, but also at a household level, in both settings. For example, a case in Cameroon, where two sisters (>50 years) one with vision impairment and the other with a severe physical impairment ran their household together. They described prioritising payment for the health needs of their brother’s six children over their own health needs. Over the last year, they had paid for three of the children to attend hospital, yet had not attended health services for themselves, even, for example, when experiencing eye pain. In India, an older woman explained that an acute health condition, such as a fever, would be immediately treated. However, despite free treatment being available, she was still waiting several years to have a second cataract treated and was dependent on family members to make the decision and accompany her:

*None of them show concern if I am laying on the bed. They will take me to the hospital when I am sick for two or three days. They will say they don’t have money, both my son and daughter-in-law say it. If you treat my eyes, I would be so grateful, …… I have blurred vision.* (India, Female, 55 years, waiting for a second cataract operation)

There also appeared to commonly be an adaptation to, and more ‘acceptance’ of chronic conditions, such as gradual loss of vision, and especially conditions that were perceived to be an inevitable part of the ageing process.

### 3.3. Type and Severity of Impairment and Interaction with the Environment

The decision to seek care also depended on the type and severity of impairment, the interaction with the physical environment, and the extent to which this then impacted on functioning and participation, and particularly in income generation. Both study settings were predominantly rural, and where subsistence farming or daily labouring was the norm, adults with musculoskeletal impairments faced greater difficulties working because of the terrain and distances they had to walk. Related to this, participants seemed to be more motivated to seek treatment for physical impairments to enable them to continue working. In contrast, people with a profound hearing impairment faced fewer difficulties with working and were seemingly less motivated to seek health/rehabilitation interventions. This was explained by one older woman with a profound hearing impairment who had not sought treatment:

*I do all my things. The only problem is that if someone is calling for me from afar, I will not be able to hear the person, except if the person comes very close and touches me........*(Female Cameroon, Profound heating impairment (Case 4803), 55 years)

For people who acquired an impairment, a very substantial proportion of our sample, the tipping point in finally deciding to seek treatment was when the impairment finally limited their ability to work. This is illustrated by an 81-year-old blind man in Cameroon, who lived with his wife and two grandchildren; his vision had deteriorated over the past 16 years but he was only now finally seeking treatment as, due to his poor vision, he could no longer work on the farm.

### 3.4. Family-Level Interaction

Despite the different social-cultural contexts, an almost universal finding from interviews in both settings was the key role of the family in decision-making about health-seeking behaviour; it was rarely the individual alone who made the decision. This was often linked to economic decision making, whereby the person responsible for the household finances made the final decision. Family members might typically be expected to come together to pay for health services:


*I went to the hospital and they gave me some eye drops.*



*And now that they ask you to come for an operation, what do you plan to do?*



*Whenever I get the money, I will go for the operation.*


Are you planning to go and have the operation soon?


*I am waiting for my children to come during this break (Christmas) and raise some money so I can go for the operation.*


(Case 171601, Male, 81 years, Blind with moderate hearing impairment, Cameroon)

The role of household members was also crucial for accompaniment to health services. This was an enabling factor to accessing health services for a number of reasons; a female might not be expected to access a service alone, an impairment might mean extra help was needed with travel or communication, or an expectation that they would need extra help in accessing a service in an unfamiliar setting. 

Overall, in Cameroon, most people described feelings supported by family members. In fact, all participants who had acquired their disability during their adult life and who had previously lived independently had returned to their families to be cared for. In contrast, in India, older people were more likely to comment on their health needs being less prioritised within the family and feelings of being a burden on the family. The older person’s heath appeared to be devalued. This was illustrated in the Indian context by an 82-year-old man who had not taken up his free eye treatment, instead, his daughter-in-law reflected the need to prioritise her treatment over his:

*My father-in-law has already reached his last days, so what is the use of doing this* [taking him for a referral]. *I want to go to a private hospital and get a check-up*. (India, Daughter-in-law with moderate visual impairment, reflecting on her father-in-law (Case 2508001).

### 3.5. Economic Factors

The financial cost of accessing treatment was a commonly identified barrier, but the issue was complex and nuanced in both settings. It included both the individual and household’s knowledge and expectations about costs for treatment, the nature of economic decision-making at the household level, and opportunity costs associated with taking time off work to seek treatment. Paradoxically, in India, although one-third of interviewees said that cost was a major barrier to accessing health services, almost every person said they would prefer to pay for private health services, rather than free government services, which was related to issues of trust, further discussed below. This was not the case in Cameroon, where people were happy to use the local government or a faith-based hospital service. However, in Cameroon, in a third of interviews, there was a low level of knowledge about the cost of treatment and a common perception that any treatment would be expensive and unaffordable:

Have you ever tried to find out how much you need to go to the hospital for your eyes to be treated?

*No, I have never found out.* (Code 320402, Male, 50 yrs., Blind, Cameroon)


*People usually tell me that even if I take 10,000frs to the hospital it will not be enough and besides I just think that the problem cannot be solved in the hospital.*


Is there any other reason why you have not gone to the hospital?

*Also, one of my sisters had the same problem and she complained to the husband, the husband said there was no money that it entails a lot to treat the ear*. (Female, 55 years, profound hearing impairment, Cameroon)

There were examples in Cameroon of people accessing community group savings schemes to pay for treatment, but this was less evident in India. Although there was a disability pension in India, it was largely targeted at those born with a disability, and few interviewees were in receipt of this pension, nor were members of disability-specific support groups.

### 3.6. Social Inclusion and Participation in the Community

Social inclusion and participation were key themes which emerged from exploring community-level factors that shaped access to health. Overall, the level of community social support and participation for the person with the disability varied between the two sites. In Cameroon, there were higher levels of participation in community life, with most interviewees providing examples of community and social support structures available to them, such as memberships of Disabled Persons Organisations (DPOs), village savings schemes and church groups. These networks also facilitated access to health services, for example, membership of village savings schemes which enabled access to funds to cover health care costs, and through memberships of DPOs, there was a greater awareness of services available, and of rights to services. In contrast, in India, the person with the disability, and particularly older people, more often expressed the view that they felt they were a burden on family members, and that their ailments were not prioritised at the household level. No one interviewed was a member of any kind of support group and there was little evidence of participation in community structures, such as local savings groups or local village committees.

### 3.7. Interaction with Service Providers

Key themes related to engagement with the health service providers were trust and acceptability; and here there was also contrast between Cameroon and India. In Cameroon, people were generally positive about the quality of the local health services, including the attitudes of staff. This was particularly the case in relation to the local faith-based hospital in the study area, which had a good reputation. In contrast, in India, the complexity of the health care provision, both government and private related, was reflected in a varied prior experience of health care provision. Trust emerged as a major issue in the India context, linked with perceptions of the service quality. Even the extremely poor were willing to take out a loan to pay for a service rather than using a free government service. This issue of trust appeared to be a key factor behind the non-uptake of the referrals for free services given during the survey in India. Lack of trust appeared to be related to perceptions of free government services being of poor quality, and/or to previous experiences of not being treated with dignity in their contact with free public health services. This was illustrated by the case of two female cousins, one with profound and another with severe hearing impairment: 

*A free service is provided, but we feel free services are not up to mark in treating the patients. So, we wish to go for paid services because paid services are worth it*. (Case 380529, India)

Interestingly, by the end of this interview, once a rapport had been developed, both women said they would be willing to take up the treatment if the interviewer was involved in the treatment:


*Are you people treating us? If there are other people treating us, then we may not visit the facility…it will be difficult to trust them. As you communicated well with us, and you know our problem, it will be easier for you to treat us, rather than a new person. You can give us good medicine.*


Further, in India, in a small number of cases, there was some concern and fear that having treatment might preclude their eligibility to the disability or pension allowance, and being able to access these was a key priority for them.

### 3.8. Referral Processes

All interviewees had received referrals to health/rehabilitation facilities as part of an earlier survey and there was commonly considerable confusion about the referral process for various reasons. People had commonly lost the referral form and did not fully understand the referral process and procedural issues, such as the time and day to attend appointments. There was still a perception by some participants that they would be required to pay for services even when they were, in fact, free, and in India, there was quite some confusion about whether transport was provided and accompaniment for the referral; there was an expectation based on prior government camps in the area, that all transport would be free.

### 3.9. Gender and Health Care-Seeking Behaviour

The intersectionality of gender and disability emerged in both sites and impacted upon care-seeking behaviour, manifest in different ways. The ‘Kom’ culture in North-West Cameroon is matriarchal, and there were examples of women with disabilities living on their own, paying for their own health treatment, and accessing village savings schemes. However, in two-thirds of interviews with females in Cameroon, and almost every interview in India, women described their dependency on male family members for financial decisions at the household level, which affected their health-seeking behaviour.

Who makes the decision about going to the hospital in the compound?


*My husband, but if he is absent, my son can take the decision.*


Who covers the cost in the hospital?

*It depends on the person that took the decision for me to go to the hospital, if it is my child [son] who took the decision that I should go to the hospital, he pays*. REF 330802 (Female, 77 yrs., Cameroon):

Who makes the decision of whether you can go to hospital?


*If my brother says, then only I will go.*


I: Which brother? You have five brothers?

R: *All brothers. If all brothers say the same thing, I will go for a check-up.* Case 3805029 25-year-old female, with a severe hearing impairment, India) 

A small number of interviews conducted with five younger women (19–35 years) born with a disability (3 India, 2 in Cameroon) illustrated their vulnerability in relation to accessing health services. Three of the five women were mothers who had been left by the father of the child and were dependent on their own family. This was illustrated in Cameroon by the case of a single woman aged 22 years, with a severe congenital physical disability. She lived with her grandmother and provided voluntary childcare for community members. Despite initial support from the Church for treatment of her impairment, her family were unable to afford a follow-up operation. She was recently diagnosed as HIV positive and ‘*pleaded*’ with her father for funds to visit the hospital. At the time of our interview, she was struggling to find funds to return for HIV treatment and to take her son for testing. Another mother, in a neighbouring village, born with a mild physical impairment, described dropping out of secondary school because of difficulty with walking. As a single parent without a livelihood, she was dependent on her father financially for accessing health services. Her main concern was her own lack of economic empowerment to access health for her child’s health and educational needs. 

### 3.10. System Level Factors

The limited availability of specialist health and rehabilitation services, such as assistive devices and physiotherapy, and the weakness of referral systems were other key barriers identified as key themes in both settings, with the exception of eye care services. In Cameroon, there was no national government training programme for any therapists at the time of the study and there were only two physical rehabilitation centres nationally. A number of ministries are involved in the provision of services for people with disabilities and a lack of integration between health and social care services offers further challenges as illustrated by the complexity of the process required to access an assistive device in Cameroon:

If a woman needs a crutch to walk, who is responsible for providing this? 

*“We (health services) are not responsible for giving her the crutch. We are responsible for assessing her. I don’t know how far they go with the specialised services and what happens next. We don’t know if social services pay for the surgery somewhere? I don’t know if they give them a monthly stipend? That I cannot tell. At my level, all I can say is that this person has this level of incapacity.”* (Senior district health staff KI01, Cameron)

In the India context, the picture was yet more complex with a plurality of health service providers available. Notions about the poor quality of services provided by the government facilities and providers, reliance on the private sector providers, lack of clear knowledge regarding the referral services and fear of losing the disability pension impacted the uptake of rehabilitation services.

## 4. Discussion

This two-country study illustrates the complex web of issues which affected the uptake of health and rehabilitation services among adults with disabilities. By comparing experiences in two very different contexts, we have drawn out similarities and differences in terms of how personal environmental factors can interact with impairments and how, in turn, this can affect uptake of health services. Key themes included individual-level factors; understanding and beliefs about an impairment/health condition, the impact of the interaction between the impairment and environment on participation and perceived urgency for treatment compared to other health conditions. At the community and household level, key themes were family dynamics and attitudes, economic factors, social inclusion and community participation. Intersectionality with gender and age were cross-cutting themes. Trust and acceptability of health service providers, lack of knowledge about the referral process and limited rehabilitation services, were the main service-level themes.

Our findings regarding the role that beliefs and understanding about an impairment play in health decision-making align with previous studies which find these factors influence the timely uptake of treatment and care [8,9,21,22,23,24]. For example, a study in Bangladesh found community and parental beliefs about their child’s disability, including whether the treatment was really necessary or whether the child “would grow out of a disability”, all influenced uptake of health-related services for the child [8]. Our study suggests additional challenges around understanding may have resulted from insufficient information given during the survey. Participants had been verbally informed about their need for services and where to seek these. However, there was still considerable confusion about the referral process and what to expect, particularly related to the costs of services. This concurs with studies in Malawi [25] and Bangladesh [8] following screening camps for children with disabilities, where confusion and misunderstanding about the referral process contributed to the non-uptake of referral. These findings highlight a critical need for more effective communication on referrals within surveys. To address this issue, a recent study in Malawi used a participatory approach to develop an intervention involving the use of counselling and information booklet [26]; this approach deserves attention within population surveys as well.

Aligning with the ICF model of disability [15], the interaction between impairments/health conditions and the context influenced the extent of participation restrictions experienced by study participants. Our study suggests that this, in turn, influenced the level of motivation to seek health services. In these rural, predominantly agricultural settings people with physical impairments appeared to face greater difficulties participating in the work activities and consequently were more motivated to seek health/rehab services compared to those with vision or hearing impairments. We describe an apparent ‘tipping point’, whereby health care seeking behaviour, for someone who acquires an impairment, is often finally initiated when a person is no longer able to work. This concurs with a study in Bangladesh which showed that treatment was sought when an impairment impacted on a person’s ability to work or conduct household activities, [27], or where when there was a loss of independence [8].

A hierarchy of health needs, both at the level of the individual and the family, was evident whereby acute health conditions were more likely to be prioritised over impairment-related services. This is perhaps unsurprising in low-income contexts where financial barriers to service uptake were evident. Further, adaptation to and ‘acceptance’ of gradual loss of functioning (e.g. vision or hearing) appeared to manifest in a lack of felt need for services—by individuals and family—especially when these conditions were perceived to be an inevitable part of ageing. This concurs with quantitative studies in LMIC which identified the lack of perceived need among older adults as a barrier to uptake of impairment related services, for example, cataract surgery [28,29]. Our findings also align with a qualitative study among older adults in Egypt and South Africa which found high levels of independence and low perceived need for intervention despite severe vision impairments [30,31]. This echoes the World Report on Ageing and Health which highlights that older people “develop their own behavioural and environmental strategies to compensate for declining function” [32]. 

Social inclusion is central to the UNCRPD and is arguably essential for achieving SDGs [2,6]. Our study demonstrates the facilitating role that inclusion in the community level, social initiatives and financial initiatives may play in health-care access; in Cameroon, engaging with the church and community groups, DPOs and village saving loan schemes enabled access to funds as well as increasing awareness of rights and availability of health services. In contrast, in India, where this community engagement did not seem to occur as much, people were more likely to report feeling a burden. These findings highlight the importance of community-based social and financial initiatives that are disability inclusive; ideally achieved by engaging with people with disabilities at all stages of planning, development and implementation.

Another key factor influencing health decision-making was the family and community context. The pivotal role of the family is similarly recognised in a limited number of previous studies on uptake of impairment related services [8,9,11,21]. For example, a study with older people in Tanzania illustrated the role of family inter-generational relationships in the low uptake of free cataract surgery [11]; younger heads of households were the decision-makers and frequently prioritised the needs of younger members. The authors offered the concept of ‘utility’ as a reason for prioritising the treatment of younger members or male members who were able to work in a wider context of difficult economic conditions and this has also been highlighted in other studies [27,31]. This concept of ‘utility’ may be a contributing factor in the current study in India, where elderly participants felt they were not prioritised for treatment within the family and instead commonly perceived themselves as a burden on the family.

Our study also highlights the intersectionality of disability with age and gender, which mirrors findings from a small number of other studies. A study with older adults in India showed that gender differences in health are likely to be modified by gender relations and kinship systems; older women with physical disabilities can be more dependent on other family members for care and support, and often have a lower status within the family, especially when widowed [33]. A study in Cameroon also indicated the particular vulnerability of women with disabilities [34]. There is a need for more nuanced research on the intersectionality of gender disability and ageing, which is context specific and takes into account kinship systems.

Economic factors are frequently cited as barriers to the uptake of health and rehabilitation services for people with disabilities services [35]. Given that people with disabilities are, on average, poorer and often have additional health care needs, they are likely to be disproportionately affected by financial barriers [36]. While financial factors were reported in both study settings, our study illustrates the complexity of issues which impact on a family’s willingness to pay for treatment. The issue of trust in the health service provider can also play an essential role in decisions to not take up a free service and willingness to pay for a private provider, which has been shown in other studies [37,38,39].

## 5. Implications

There are a number of implications and recommendations arising from these findings. Firstly, this research suggests that more chronic health conditions and impairments which have a more gradual impact on functioning may require a different health promotion strategy compared to acute conditions to stimulate the timely uptake of treatment. Second, any approach also needs to target the family, not just the individual, recognising family-level decision making processes and their role in facilitating access to treatment. Third, given the heterogeneity, a more nuanced approach to planning inclusive health services is required, moving away from an approach that ‘one size fits all’, for people with disabilities. This includes a better understanding of intersectionality in the context of health-seeking, for example, of disability and gender and disability and age. Fourth, communication about referral pathways needs to be improved in the context of population surveys. Additional investments in counselling about the condition and benefits of rehabilitation to the individual and family members coupled with a more focussed follow up may be important to facilitate referral uptake.

## 6. Strengths and Limitations

This study had several strengths. Qualitative data collection enabled an in-depth exploration of factors influencing health-seeking from two contrasting settings. We interviewed people with a range of impairment types and severity in order to explore different perspectives and reflecting the heterogeneity amongst people with disabilities. There were also limitations. Four people with a severe hearing impairment could not be directly interviewed due to lack of sign-language and instead, interviews were conducted with a primary caregiver which is likely to have brought a different perspective. Further, we did not include people with intellectual impairment or with mental health conditions in this study, and this should be the focus of future research on this subject. Another potential limitation was the role of the political context in India at the time of the field work when an increase in disability allowance was a possible outcome of the State election and concerns among participants about their eligibility to this improved allowance may have affected their responses. There is also a need to have more policy and system-level research to better understand barriers on the supply side, in order to provide a more coherent picture of both demand and supply-side factors. This should include an understanding of the role that the NGO sector, including faith-based organisations, play, given they are often a key player in the provision of health rehabilitation services in many low-income settings.

## 7. Conclusions

This study illustrates the importance of understanding the lived realities of people’s lives when addressing the right of people with disabilities to access the highest standard of healthcare. In line with the ICF, the interaction of environmental and personal factors with the impairment, and their levels of participation and inclusion in communities, all contribute to decisions to take up services. To ensure that people with disabilities are not left behind, this study highlights the need for a multi-faceted and contextual response, applying a gender lens.

## Figures and Tables

**Table 1 ijerph-16-01126-t001:** The samples for Cameroon and India.

Country/Impairment Type			Male			Female	
**Cameroon**							
		19–35	36–60	61+	19–35	36–60	61+
Hearing	Moderate			1	1	2	1
	Severe						2
Vision	Moderate	1	1	1		1	
	Severe		2	3			
Musculoskeletal	Moderate	1			1	2	1
	Severe	3	2	1	3		
**India ***							
Hearing	Moderate		1	1			1
	Severe		2		3	2	
Vision	Moderate		1			1	
	Severe	1	1	1			1
Musculoskeletal	Moderate	2	1				
	Severe	1	1			1	
Multiple	moderate						
	severe *		1		1	2	1
Totals		9	13	8	9	11	7

* India sample included 4 interviewees only identified as a mild disability with Washington Group questions.

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
