# Peer review of "Barriers and Facilitators to Accessing Health Services: A Qualitative Study Amongst People with Disabilities in Cameroon and India"

_ijerph, 2019, doi:10.3390/ijerph16071126_

Round 1

Reviewer 1 Report

I believe that the article improves with the new details introduced.

The authors followed the indications made in the review.

Author Response

Reviewer 1: Happy with all the amendments

We thank the reviewer for this positive feedback

Reviewer 2 Report

Some essential background information such as population in the catchment area, particularly disabled populations; number of persons in a typical household, and life expectancy etc. The lines only told of the absence of rehabilitation service in government facilities and there is no mention if other charities are providing services to the disabled. 

the quote of the elderly case that he is in his "last days" - how old is the case and is it typical or atypical for that age to be perceived as undervalued? Some clarification is needed.

The claim that gender and age issues cut across themes, would require the illustration on how gender and age really affect many different themes. At the moment, it is only discussed generally. I do not see the gender and age issues complicate different themes..

The interactionist perspective of ICD does not stand out in the result nor the discussion. The findings are put in personal and environment layers and I do not read much how they  transact with each other.

Author Response

Reviewer 2:

Some essential background information such as population in the catchment area, particularly disabled populations; number of persons in a typical household, and life expectancy etc.

We have now added more specific background data:

(1)district population size, (2) prevalence of disability, (3) proportion rural (4)  Human Development Index Ranking and (5) life expectancy for the country. See line 103- 138:

-             

The lines only told of the absence of rehabilitation service in government facilities and there is no mention if other charities are providing services to the disabled.

We do agree that in many low-income countries, charities, INGOs, and faith-based organisations provide important services for people with disabilities.  We do outline that in the Cameroon context, and have added  an extra sentence on the provision of assistive devices through the NGO sector.  This now reads:

Line 109: At each level there are a mix of government and private health facilities. There are no rehabilitation services in the government health facilities; there is one large faith-based hospital which offers rehabilitation services, and a number of NGOs also provide support, which includes providing assistive devices.

In India the role of the NGO sector is referred to (line 134):

Rehabilitation services are mostly provided by the government systems supported by the NGO sector, whilst Certification and grading of disability is done through the government health system.

In the strengths and limitations section we also return to the role of faith-based organisations and have also expanded this to include the NGO sector more generally. It now reads Line 699:

There is also a need to have more policy and system-level research to better understand barriers on the supply side, in order to provide a more coherent picture of both demand and supply-side factors. This should include an understanding of the  role that the NGO sector, including faith-based organisations play, given they are often a key player in the provision of health rehabilitation services in many low-income settings.

the quote of the elderly case that he is in his "last days" - how old is the case and is it typical or atypical for that age to be perceived as undervalued? Some clarification is needed.

In the text preceding the quote we already include the age (82 years) of the father-in-law- see line 357.  We have now added this to the end of the quote for further clarity.

While we did  not explicitly explore attitudes towards older adults, and therefore are limited in the generalisations were can make, several interviews suggested that their health is not prioritised or valued in the same way as younger adults. We already highlight this in the discussion in exploring factors which impact on family-level decision making. See line 685 onwards. Also in  response to the previous round of reviewers comments, we added a paragraph in  the discussion exploring further the findings on aging and gender and disability (line 697) and calling for ‘a more nuanced research on the intersectionality of  gender disability and aging, which is context specific, and takes into account kinship systems.’ (line 700).

The claim that gender and age issues cut across themes, would require the illustration on how gender and age really affect many different themes. At the moment, it is only discussed generally. I do not see the gender and age issues complicate different themes.

We have now removed the comment about gender as a cross cutting issue, and instead have focussed this paragraph  on the issue of the intersectionality of disability with age and gender only.

The interactionist perspective of ICD does not stand out in the result nor the discussion. The findings are put in personal and environment layers and I do not read much how they  transact with each other.

We outline in this study that we draw upon two conceptual frameworks. The main framework we used is the socio-ecological framework.  The section on the conceptual framework now reads

 Firstly, the socio-ecological framework [1, 2] outlines the  interactive effects of personal and environmental factors at different levels: individual, interpersonal,  community and organisational.  In this study, the emphasis was at the level of micro-system; with a focus on the individual and their perspective on their interaction with the family and community. Secondly, we draw upon  the World Health Organisations International Classification of Functioning, Disability and Health framework [3]

 In response to the last set of reviews, we ensured that the way results are presented is more clearly  aligned to this framework. However, we do also draw upon the ICD. We highlight in several places in the results how the interaction between a health condition with the environment, impacts on activities and participation,  (for example from Line 304 and from Line 400).  We also specifically comment on how our findings align with the ICF model, and specifically the interaction with the environment (see lines 571 , 599, 608) and how this influences health seeking behaviour.

Reviewer 3 Report

This paper has been revised appropriately. One more suggestion is that the author should consider there are only 30 informants will include all the different kinds of disabilities, may be it can have a study limitation discuss in the content. Another point is the multiple disabilities cases include mental impairments? and how to interview these cases? the author should describe the process.

Author Response

Reviewer 3

This paper has been revised appropriately. One more suggestion is that the author should consider there are only 30 informants will include all the different kinds of disabilities, may be it can have a study limitation discuss in the content. Another point is the multiple disabilities cases include mental impairments? and how to interview these cases? the author should describe the process.

We thank the reviewer for their positive feedback.  We included people with a range of sensory (vision, hearing) and physical disabilities of varying severity and feel the heterogeneity of the sample in this regard was a strength of the project.  However, we did not include people with a mental disorders/intellectual disability. This is already stated in the methods and in the limitations (lines 168). The point raised by the reviewer about challenges conducting direct interviews was, therefore, not an issue in this study.  However, we do recognise that not including people with mental/intellectual disability was also a study limitation and had already stated this in the limitation section of the discussion (lines 691):

This manuscript is a resubmission of an earlier submission. The following is a list of the peer review reports and author responses from that submission.

Round 1

Reviewer 1 Report

In the introduction, it would be good to report on the structure of health services in both countries and normal access to these services. This would place the context of the study.

The sample is well balanced.

The gender perspective helps to know the differences in access to health services and this is very relevant.

It could also have been interesting to explore the religious factor as a determinant in access to health services. Also the role played by religious organizations in this matter.

Author Response

Response to Reviewer 1

Comments and Suggestions for Authors

In the introduction, it would be good to report on the structure of health services in both countries and normal access to these services. This would place the context of the study.

Thank you for this suggestion. We have added further details on both countries as follows

Added material for Cameroon Context ( line 108):

The health system is pyramidal with a central, regional and district level of services, and at each level there is a mix of government and private health facilities There are no rehabilitation services in government health facilities. . In the Fundong district there is a large faith-based hospital which offers rehabilitation services that are unique, and as a consequence they cover the entire region as well as  neighbouring regions.

Added material for the India Context (line 129):

Healthcare is a mixed services model as there is no national health service. There is a socio-economic gradient with the rich preferring private health care and the poor seeking public health systems. Usually the more deprived areas depend on public health systems. However, a significant proportion of people still depend on village level health practitioners .Rehabilitation services are mostly provided by the government systems supported by the NGO sector,  wherever they are available. Certification and grading of disability is done by the government health system

The sample is well balanced.

The gender perspective helps to know the differences in access to health services and this is very relevant.

It could also have been interesting to explore the religious factor as a determinant in access to health services. Also the role played by religious organizations in this matter.

We agree this is a very interesting area that deserves further attention, . We explore make reference to role of religion/religious organisations in the section about membership of community organisations, which includes membership of the church within the Cameroon context. Further information was not available in our data to allow us to explore this in more depth. However, we have added a sentence within the limitations section (in the discussion) to highlight this is an area which deserves further research..

Line 721: Finally, there is also a need to have more policy and system-level research to better understand barriers on the supply side. This should include the role that religion might play in these contexts,  including the role of faith-based health services play in uptake of health services, given they are often a key player in the provision of health rehabilitation services in many low-income settings.

Reviewer 2 Report

Qualitative inquiry into public health issues of disabled population is rare and valuable. Yet there are places for improvement before the qualitative paper is at a publishable standard.

1.       There has been no justification given on why two different places, one in Africa and one in India are chosen and contrasted. Although the fact of disability and classification may be the same/standardized across countries, the meaning of disability as a whole, and specific disability in particular, the context of services setup, and accessibility issues can be highly contextual.

2.       The explanation of theoretical framework is very brief and it is unclear how the framework led to the design of the interview questions. In the analysis, it is noticed that organizational read-tap, bureaucracy of health facilities, half-hearted government policy and corruption issues which are commonly found in developing countries have not been touched upon. These are all about the context of one’s uptake (or not uptake) of service.

3.       The authors cited some of the major international policy papers that state the ideal of health equity for the disabled, yet it is quite clear that in many countries it is just paperwork rather than genuine effort to close the discrepancy between what is desired and what is. Lived realities have very little to do with these documents. When the author commented India and Cameroon as being fragmented in services provision and a lack of integration between health and social care, it appears to be high-sounding.

4.       The sample subjects included two types of sensory disabilities (hearing and visual impairment) and one physical disability, and excluded intellectual disability and mental illness. The selection appears to be deliberate but there is no clear justification. It might be “purer” and more indepth if the paper focused on a particular group like physically challenged or sensory impaired rather than a mix of them; or one countries instead of two. Secondly, the study also interviewed some “informants” who are basically service providers. it is unclear their views have already been put in the text or whether the narratives are only from the disabled persons.

5.       The background details of the subjects is shown only by disability type, severity, broad age group and gender. Disability coping is very much related to internal (familial) and external resources. There is a need to indicate the socio-demographic listing of the participants. 

6.       There are places with unclear meaning.

a.       P.4 line 159-161 “Further, factors which impacted upon the specific uptake of services, which participants had been referred to as part of the disability survey, were interwoven into the narratives, and were not always clearly delineated.”

b.      P.5 line 185 What do you mean by “hierarchy of impairments and illness”? You actually illustrated a lower priority of the need of a disabled elderly person, in the presence of a competing health needs from six children.

c.       P.6 line 228-230, it is mentioned the “tipping point” condition when the productivity is affected motivated some to access treatment. This obviously does not make sense for those with congenital problem or disability acquired during very early years and before the working years.

d.      P.6 quoted the example of an Indian old man who di not take up his free eye treatment, yet the daughter-in-law wanted to go instead to a private hospital for a service. I am not sure whether authors wanted to say elderly are valued or de-valued.

e.       P.7; It is unclear what authors wanted to bring out about the section “gender and healthcare”. There is no doubt that patriarchal societies always put women in a disadvantaged position in terms of resources usage, and women are more likely than men to become the primary family caregivers for the disabled, ill and old. I am uncertain when the authors referred to the gender of the disabled persons, or the gender of the caregivers, or the interaction between the two. The paper claimed to have a gender len, but I do not think it has been brought out clearly.

f.        The paper has brought out the issue of trust but it can be a result as well as a cause for not taking up the healthcare. There is no elaboration on why some healthcare is not trust-worthy enough.

7.       The findings spread over different levels and it may be helpful to summarize the points if the authors can put the key findings into conceptual maps or framed by the theoretical framework.

Author Response

Reviewer 2:

Open Review

Qualitative inquiry into public health issues of disabled population is rare and valuable. Yet there are places for improvement before the qualitative paper is at a publishable standard.

1.     There has been no justification given on why two different places, one in Africa and one in India are chosen and contrasted. Although the fact of disability and classification may be the same/standardized across countries, the meaning of disability as a whole, and specific disability in particular, the context of services setup, and accessibility issues can be highly contextual.

We agree that disability and accessibility issues are highly contextual and feel that highlighting this is one of the merits of this two-country comparative study. Through comparing and contrasting two different countries from different regions we aimed to highlight common similarities and differences. In terms of reasons for the selection of these settings - this qualitative study built on a wider study which sought to provide evidence on the dynamics of living with disabilities in two contrasting lower middle income settings, one in Africa and one in Asia. Site selection within each country was determined by availability of onward referral pathways for health and rehabilitation services, and in accordance with strong research partner availability.

To this effect we have added an additional line of explanation and one further reference of the original survey at line 139, plus an additional reference to this study: Mactaggart, I., Kuper, H., Murthy, G. V. S., Sagar, J., Oye, J., & Polack, S. (2016). Assessing health and rehabilitation needs of people with disabilities in Cameroon and India. Disability and rehabilitation38(18), 1757-1764

This study built on earlier large-scale surveys which aimed to assess the prevalence and impact of disability in two contrasting LMIC settings; one African and one Asian setting. This study built on an earlier large- scale survey to assess health and rehabilitation needs for people living with disabilities in one African and one Asian setting. The methodology and results of these surveys are available in detail elsewhere[3, 17].

2.     The explanation of theoretical framework is very brief and it is unclear how the framework led to the design of the interview questions. In the analysis, it is noticed that organizational read-tap, bureaucracy of health facilities, half-hearted government policy and corruption issues which are commonly found in developing countries have not been touched upon. These are all about the context of one’s uptake (or not uptake) of service.

We agree that the use of the socio-ecological framework suggests detailed exploration at all levels from the micro-system through to the macro-system.  Whereas, in fact, the emphasis of this research was focussed at the level of the micro-and mesosystem, that is,  focussing on the perspective of the  individual, and their interaction with their family and community, including services, and how this then impacted upon access to a health. We have clarified this accordingly :

Line 96:

Firstly, the socio-ecological framework [13, 14] outlines the  interactive effects of personal and environmental factors at different levels: individual, interpersonal,  community and organisational.  In this study we  this framework to guide the analysis in terms of understanding factors at the level of micro-system (individual) and meso-system (their environment and  interaction with the family and wider community, including services).

.

3.     The authors cited some of the major international policy papers that state the ideal of health equity for the disabled, yet it is quite clear that in many countries it is just paperwork rather than genuine effort to close the discrepancy between what is desired and what is. Lived realities have very little to do with these documents. When the author commented India and Cameroon as being fragmented in services provision and a lack of integration between health and social care, it appears to be high-sounding.

We were not entirely clear of the point that the reviewer was making, but interpreted that the need for more explanation of this topic about fragmentation of services and poor integration; all important supply-side issues.    As explained in response to point 2, the main focus of the study was on the individual perspectives, demand-side barriers, and capturing the voice of the person with a disability and their narrative and we hope we have made this clearer now. We have now added a section to the introduction (as above Line 96)  to emphasise that this is the focus of the article. However, we did capture some broader contextual information from key informant interviews on the barriers around service delivery, which we chose to include to give more context.    Further, we have added a sentence to the Discussion Session  on the need for more policy and system-level research to understand the barriers on the supply side.

See additional text in the discussion (Line 747)

There is also a need to have more policy and system-level research to better understand barriers on the supply side, in order to provide a more coherent picture of both demand and supply-side factors.

4.     The sample subjects included two types of sensory disabilities (hearing and visual impairment) and one physical disability, and excluded intellectual disability and mental illness. The selection appears to be deliberate but there is no clear justification. It might be “purer” and more indepth if the paper focused on a particular group like physically challenged or sensory impaired rather than a mix of them; or one countries instead of two. Secondly, the study also interviewed some “informants” who are basically service providers. it is unclear their views have already been put in the text or whether the narratives are only from the disabled persons.

The original prevalence survey included a self-reported tool (the Washington Group Extended Set) that captured some limited information on reported functional limitations related to remembering/concentrating or communicating and anxiety/depression. However, these group were not included in the qualitative component. This was, in part, because, as the reviewer points out including too many different groups can limit depth, and also because we wanted to understand poor uptake of referrals to existing services in the district and for these target groups there were very few services available.  We have added a comment that a limitation of this study was that it did not include intellectual disability or mental illness and that this should be addressed in future research

There are of advantages and disadvantages to taking a broader approach which includes different types and severity of disability.  Although it might be ‘purer’ to focus on one disability type only, given that the qualitative built on a large disability survey, we were in a good position to easily select people with a range of different impairments.  By comparing across these three  impairments, also enabled us to explore how environmental factors interact differently with different impairments and then shape the experience of disability. This is illuminated in the section on ‘Type and severity of impairment and interaction with the environment’.

We have added a sentence to our discussion to further highlight this.

Line 475  Discussion. This two-country study illustrates the complex web of issues which affect uptake of health and rehabilitation services among adults with disabilities. By comparing experiences in two very different contexts, we have been able to draw out similarities and  differences in terms of how environmental factors interact with the impairment, and in turn, how this affects uptake of health services

5.     The background details of the subjects is shown only by disability type, severity, broad age group and gender. Disability coping is very much related to internal (familial) and external resources. There is a need to indicate the socio-demographic listing of the participants. 

We have now included details on education level and work status of the individuals included at line 183:

Overall levels of education were low; 43 % with no education, 47% primary education and 3% with secondary education  in Cameroon, and  65% with  no education, 13% primary level, and 23% secondary education in India. 67% of the Cameroon sample had worked in the last 12 months, and 61 % in India.  

6.       P.4 line 159-161 “Further, factors which impacted upon the specific uptake of services, which participants had been referred to as part of the disability survey, were interwoven into the narratives, and were not always clearly delineated.”

Further, the factors that influenced uptake of those impairment related services that participants were referred to in the survey were interwoven into their narratives about access to health care more generally, as opposed to the two being clearly delineated.

b.       P.6 line 228-230, it is mentioned the “tipping point” condition when the productivity is affected motivated some to access treatment. This obviously does not make sense for those with congenital problem or disability acquired during very early years and before the working years.

> This is an important point, and we have amended the text for clarity in line 314

For people who acquired an impairment, which was a substantial proportion of our sample,  the tipping point in finally deciding to seek treatment was when the  impairment finally limited their ability to work.

d.       P.7; It is unclear what authors wanted to bring out about the section “gender and healthcare”. There is no doubt that patriarchal societies always put women in a disadvantaged position in terms of resources usage, and women are more likely than men to become the primary family caregivers for the disabled, ill and old. I am uncertain when the authors referred to the gender of the disabled persons, or the gender of the caregivers, or the interaction between the two. The paper claimed to have a gender len, but I do not think it has been brought out clearly.

Many thanks for this feedback, Whilst we recognise that gender and caregiving is a major issue which is under-recognised, this study was about adults living with disabilities. Therefore, our the focus with regards to gender was on its intersectionality with disability.

This study was not explicitly focussed on gender and disability, but the data suggested this is an important issue which deserves more attention. We have expanded our discussion, drawing on relevant literature and highlighting,that this should be the focus of future research:

Additional references: (Kiani, S., Women with disabilities in the North West province of Cameroon: resilient and deserving of greater attention. Disability & Society, 2009. 24(4): p. 517-531, and Sengupta, M. and E.M. Agree, Gender and disability among older adults in North and South India: differences associated with coresidence and marriage. Journal of cross-cultural gerontology, 2002. 17(4): p. 313-336.)  

Line 674: Gender was a cross-cutting issue across both socio-cultural settings, impacting on the uptake of health services. The intersectionality of disability with gender, and aging,  aligns with another study of older adults in India which showed that gender differences in health are likely to be modified by gender relations and kinship systems. The authors suggest that older women with physical disabilities in North India, especially when widowed, have a lower status within the family, and yet are dependent on other family members for care and support  [32].  A study in Cameroon also indicated the particular vulnerability of women with disabilities [33]. There is a need for more nuanced research on the intersectionality of  gender disability and aging, which is context specific., and takes into account kinship systems.

f.       The findings spread over different levels and it may be helpful to summarize the points if the authors can put the key findings into conceptual maps or framed by the theoretical framework.

Thank you for this suggestion. We have re-named some of the headings in the findings so they are more clearly aligned to different layers of the socio-ecological framework; results at the level of (1) the Individual , (2) individual’s interaction with the family and (3) interaction with the community, with gender as a cross-cutting issue.  This also necessitated some re-ordering of the results to link into the layers of the socio-economic framework, in response to the reviewer’s comments on the use of the conceptual framework.